# SLEC: A Novel Serverless RFID Authentication Protocol Based on Elliptic Curve Cryptography

**Rania Baashirah [1],* and Abdelshakour Abuzneid [2],*** 

1   Department of Software Engineering, College of Engineering, University of Business and Technology, Jeddah, Saudi Arabia
2   Department of Computer Science and Engineering, School of Engineering, University of Bridgeport, Bridgeport, CT 06604, USA
*   Correspondence: r.baashirah@ubt.edu.sa (R.B.); abuzneid@bridgeport.edu (A.A.); Tel.: +1-(203)-576-4113  (A.A.)

**Abstract:** Internet of Things (IoT) is a new paradigm that has been evolving into the wireless sensor networks to expand the scope of networked devices (or things). This evolution drives communication engineers to design secure and reliable communication at a low cost for many network applications such as radio frequency identification (RFID). In the RFID system, servers, readers, and tags communicate wirelessly. Therefore, mutual authentication is necessary to ensure secure communication. Normally, a central server supports the authentication of readers and tags by distributing and managing the credentials. Recent lightweight RFID authentication protocols have been proposed to satisfy the security features of RFID networks. Using a serverless RFID system is an alternative solution to using a central server. In this model, both the reader and the tag perform mutual authentication without the need for the central server. However, many security challenges arise from implementing lightweight authentication protocols in serverless RFID systems. We propose a new secure serverless RFID authentication protocol based on the famous elliptic curve cryptography (ECC). The protocol also maintains the confidentiality and privacy of the messages, tag information, and location. Although most of the current serverless protocols assume secure channels in the setup phase, we assume an insecure environment during the setup phase between the servers, readers, and tags. We ensure that the credentials can be renewed by any checkpoint server in the mobile RFID network. Thus, we implement ECC in the setup phase (renewal phase), to transmit and store the communication credentials of the server to multiple readers so that the tags can perform the mutual authentication successfully while far from the server. The proposed protocol is compared with other serverless frameworks proposed in the literature in terms of computation cost and attacks resistance.

**Keywords:** RFID; serverless; mutual authentication; IoT; ECC

## 1. Introduction

Radio frequency identification (RFID) is a technology that allows us to detect items through electromagnetic waves. Many existing industries improve their daily business operations and annual profits by using RFID systems and applications. It involves product allocation, supply chain management, inventory tracking, toll collections, access control, and more. Furthermore, RFID technology has been expanding significantly to become essential for daily life matters and integrated with household, automotive, and smartphone applications.

A simple RFID system consists of three main components: tags, readers, and servers [1]. The RFID tag is a small chip that works as a transponder to a query signal. It is usually attached to an item to be detected among many other tagged items in the same network. The tag is composed of a small

antenna that is attached to a microchip and a small memory to store the identity and secret credentials of the object.

On the other hand, a reader is a receiver that works as a scanner to collect data from the tag. It is placed in either a fixed or mobile location to interrogate the tags in the surrounding field. The server is a data processing unit that stores, controls, and manages the data used during the communication between the reader and the tag. An RFID system is depicted in Figure 1 [2]. Because the communication channel between the reader and the tag is assumed to be insecure while messages are transmitted, the information would be vulnerable to security attacks such as replay attack, impersonation, traceability, man-in-the-middle, desynchronization, denial of service, cloning, and disclosure attack. A secure RFID system must be able to resist different types of attacks through maintaining system requirements of mutual authentication, confidentiality, integrity, availability, privacy, and forward and backward secrecy.

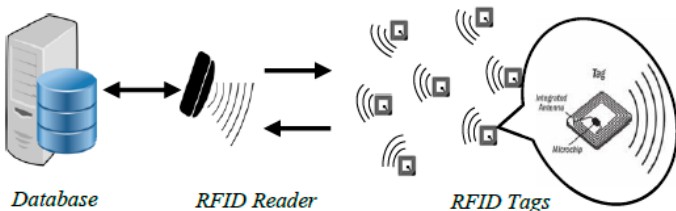

**Figure 1.** Basic radio frequency identification (RFID) network.

Identifying products and humans and authenticating their validity is a crucial daily matter, especially in mobile RFID systems, where the readers and tags exist in locations far from the central server. In situations such as the cars dealership industry, a large number of vehicles needs to be identified and located off the dealership facilities and during transfer between locations. Identifying an asset starts by authenticating the real asset among many other assets. This is done using a secure authentication between the tag and the reader to identify the vehicle and register the location of the legitimate car. Tracking vehicles could use the owners' information to manage the transportation and avoid unwanted incidents such as car theft or location lost. Authorized facilities must control access to the cars' information, by allowing only authorized users to access the data. Individuals' privacy is essential, so any adversary should not be able to obtain any valuable information about the vehicles from the attached tags. Privacy can be achieved by confusing the adversary by sending noise signals from different locations to avoid capturing the data and prevent tracing back the transmitted signals using network traffic analysis.

The continual development of the RFID system led to the introduction of the concept of serverless RFID, where the communication between the RFID reader and tag does not involve a central server. This becomes a necessity if the objects are mobile and destined from the server. This innovative scheme raised significant security issues in the RFID system because both the reader and tag should form an autonomous communication venue. That means authentication and encryption needs to be done by the tags as well as the readers, which is intensive computing. Multiple serverless RFID protocols have been proposed using lightweight operations, such as pseudo-random number generator and exclusive-OR operations [3]. Even though these protocols meet the requirements of the RFID passive tags limited resources, they are still exposed to security breach due to the lightweight operations used primarily in the reader and tag for authentication.

Many of the currently proposed RFID authentication protocols assume a secure environment during the setup phase between the server, the reader, and the tags, which is not realistic in most of the cases. In our work, we provide a security mechanism to the setup phase. The setup phase updates the protocol parameters to perform asset authentication. The process misleads the adversary about the location of the asset. The proposed Serverless RFID Authentication Protocol Based on Elliptic Curve Cryptography (SLEC) gives the reader the ability to handle signals that are sent from a group

of tags at the time, and helps to mislead the adversary from tracing the real tags. We also secure the tag information using keys shared between the reader and tag. ECC public-key algorithm is used and compared with other ECC protocols to validate the proposed model. A few analysis models are developed to prove the novelty of the proposed work. In Section 2, we discuss related work that we found in the literature. In Sections 3 and 4, we discuss the system model and SLEC protocol details. In Sections 5 and 6, we present the security analysis and formal verification test. We conclude our work in Section 7.

## 2. Related Work

In this section, we present some of the previous work on the RFID authentication protocol that is particularly based on the mutual authentication between the reader and the tag to establish a secure communication in the RFID network. The low-cost passive RFID tag is small chip designed with limited resources that are only able to perform low computations. Consequently, a literature review was conducted by Baashirah et al. [1] to classify the RFID authentication protocols based on the algorithm complexity that the tag has to perform for its responses. This classification includes: heavy-weight, simple-weight, light-weight, and ultra-lightweight. According to Baashirah et al., the heavy-weight protocols use heavy computation algorithms, such as public key and symmetric key cryptography. Their computation overhead is beyond the ability of the RFID passive tags to process. The simple-weight protocols use a simple operation such as the hash function that generates a significant computation overhead on the tag due to the limited tag resources. The light-weight protocols use operation with low computation overhead as oneway hash function, cyclic redundancy check (CRC), and pseudo-random number generator (PRNG). Lastly, the ultra-lightweight protocols use operations with considerably low computation overhead, such as simple bitwise operations that can be performed at a minimal cost possible for the tag.

### 2.1. Heavyweight Protocols

Wang and Sarma [4] proposed two session-based authentication protocols, SB-A and SB-B, for the reader–tag authentication based on symmetric key encryption. Their protocols are based on a symmetric cryptography algorithm to provide low-cost authentication using two types of passive tags to ensure privacy and access control. However, symmetric cryptography algorithms such as Data Encryption Standard (DES) and Advanced Encryption Standard (AES) are expensive operations that require a lot of computational overhead on the tag.

Considering highlevel security, the elliptic curve cryptography is introduced as one of the public-key schemes for low constraint devices. Using the same concept of public and private keys for both the reader and the tag, the RFID system is able to defend most of its security threats. Ryu et al. [5] proposed elliptic curve cryptography-based untraceable authentication protocol (ECU) using the Schnorr signature scheme to overcome the traceability problem. ECU protocol is proposed to overcome the issues of three recent untraceable authentication protocols based on elliptic curve algorithms: strong privacy-preserving authentication protocol (SPA) [6], efficient mutual authentication protocol (EMA) [7], and ECC-based authentication protocol PII [8]. Ryu's protocol generates a digital signature with an appendix on the binary message of arbitrary length and requires a cryptographic hash function. The sender's session key is combined with the receiver's public key to provide privacy, in which the message can be verified by only the receiver's private key. Ryu's protocol is secure against replay attacks, impersonate attacks, traceability attacks, and it maintains forward security. However, ECU requires two scalar multiplications, two hash functions, a message total size of 544 bits, and two communications between tag and reader, which creates a heavy computation overhead on the tag. Even though this protocol requires complex computations associated with scalar multiplications and a hash function, it does not authenticate the reader.

To reduce the tag's overhead in heavyweight protocols, Yao et al. [9] introduced the reviving-under-dos (RUND) authentication protocol to defend against denial of service (DoS) and

preserve user privacy by powering up the tag to do complex computing for symmetric and public-key cryptography. The more signals there are in communication, the more power charges the tag. Even though the overall efficiency of RUND is O(1), it is still not compliant with the Electronic Product Code Class1 Generation2 (EPC C1 G2) standard [10], which is defined by EPCGlobal Inc. for RFID data communication.

## 2.2. Simple-Weight Protocols

To better improve the performance of RFID protocols and reduce the power that is needed for complex operations in ECC-based protocols, Farash [11] proposed a mutual authentication protocol (IECC) based on the elliptic curve to overcome the vulnerabilities in Chou's authentication protocol (EMA) [7]. The computation cost of IECC still needs to be reduced for practical implementation.

Zhang and Qi [12] also proposed another ECC based authentication protocol (EECC) to withstand the security weaknesses of Chou's protocol, EMA [7]. In comparison to EMA protocol, EECC protocol resulted in better performance and security resistance to impersonate and forward security attacks. However, Baashirah et al. [13] found that Zhang and Qi's protocol is vulnerable to forward traceability and reader impersonate attack, as an adversary can compromise the private key of the reader by obtaining the tag's secret identifier.

Baashirah et al. improved Zhang and Qi's (EECC) protocol and proposed Hash-Based Authentication Protocol using Elliptic Curve protocol (HBEC) protocol that is based on securing the tag's secret identifier using a oneway hash function. The HBEC protocol overcomes the security flaws in EECC protocol to provide high security, even though the extra hash function adds more overhead to the computation, which should be addressed for the network scalability.

B.Chen [14] proposed a role-based access control (RBAC) protocol for mobile RFID, to enable user privacy, role, and access control through the back-end server based on a certification mechanism. RBAC assigns role classes as keys to control the information and the number of times each reader can read a tag. The protocol is effective against traceability, replay attack, unauthorized access, and integrity. However, RBAC uses one encryption mechanism that is excessive for low-cost passive tags.

## 2.3. Lightweight Protocols

Successful businesses require a secure RFID system that is able to perform efficiently, using low computation overhead at the minimal cost possible. The prominent RFID protocols employ low-cost operations that are handled by the inexpensive passive tags for practical applications. Accordingly, an efficient authentication protocol (SEAS) was proposed by Dass and Om [15], which uses lightweight operations and a pseudo-random number generator (PRNG) for low computations on the tag.

An alternative solution to replace the central database in the RFID system is to use a *serverless* model in which the database server does not maintain a connection with the readers and tags during the communication. Regarding this challenge, Mtita et al. [16] proposed (SAP), a serverless security protocol used for the mass authentication of RFID tags in the presence of untrusted readers. The protocol has been proven using the *CryptoVerif* tool [17], which was shown to have low computation overhead and resources.

To reduce the communication time during the authentication session, K. Lee et al. [18] proposed the efficient passively-untraceable authentication protocol (EP-UAP) based on randomized hash-lock protocol. The system precomputes all of the necessary computations before the system initialization in EP-UAP, so only low computation overhead is required on the tag side during the process phase. Because precomputing the random responses requires a storage memory in the database, EP-UAP is preferred for small to medium networks, as the storage memory increases when the number of tags increases. The protocol shows considerable improvement over the randomized hash-lock protocol in terms of computation time. However, it requires 100 MB of database storage memory in the database, and it cannot defend active attacks such as impersonate and replay attacks since the random responses depend on the database/reader.

## 2.4. Ultra-Lightweight Protocols

As mentioned earlier in this paper, passive tags are small chips with scarce resources that can only support low-cost operations. The goal of ultra-lightweight protocols is to reduce the cost of RFID systems to a minimum, and provide strong security for promising future use. In this regard, Sundaresan et al. [3] introduced an ultra-lightweight serverless protocol (STS) using only simple XOR and 128-bit PRNG operations that require less than 2000 gates to conform to EPC C1 G2 Standard. In STS protocol, the tag can preserve its location privacy by responding as a noise tag, and the pseudo-random numbers that are used in communication between readers and tags are made hidden using a blind factor to overcome impersonation attacks.

Aggarwal and Das [19] proposed the CWH+ protocol, which is based on a previous version introduced by Y. Chen, Wang, and Hwang (CWH) [20]. The CWH+ protocol solves the problem of full disclosure attack due to the simple XOR operation that is used in the authentication message, which uses the bit rotation and shifting operation on the message before transmission to increase the protocol complexity. CWH+ protocol is resistant to replay attack, forge attack, and DoS with a very efficient computation.

Huang and Li [21] proposed and implemented two improved protocols of RFID based on generating the PadGen function in the ISO 18000-6C [22] protocol to protect the memory with a 32-bit access password (Apwd) before transmitting the data. They used PadGen with XOR in one protocol (PGX) and PadGen with Modulo operation (PGM) in the other one. Even though both schemes conform to the EPC C1 G2 standard, the security level of the PGM scheme is higher because of the low-cost implementation, but the computation cost in PGX is lower.

Based on the related work and its implications, it is noted that the heavyweight and simple-weight protocols are not feasible for practical implementation. On the other hand, lightweight and ultra-lightweight protocols use only simple operations within the tag computation limits and show the lowest tag computation overhead level, so they are most used in the current applications. Many RFID protocols are proposed to defend against different attacks. However, several vulnerabilities are detected in the lightweight protocols because it is easy to break out the security of their simple operations.

## 3. System Model

In the event of a mobile RFID system, a reader and a tag start communicating by authenticating each other without a central database to perform the necessary calculations to establish a secure communication channel. In the authentication session, the reader and tag transmit challenging messages that can only be computed and verified by a legitimate entity. The transmitted messages should be confidential; they require encryption and decryption using low-cost operations within the ability of the passive tag to process. The privacy of the tag is also required to protect the tag secret information and location from being exposed to adversaries. Because the secure algorithms require extensive computations, it is important to minimize the communication signals in the network especially when the number of tags is high. We developed a secure and appropriate authentication algorithm that maintains the security of the system and privacy of the tags, while minimizing the communication signals in the network to reduce the computation overhead on both the reader and the tag. In this section, we present the system model and the communication model of the proposed work.

### 3.1. Network Model

A number of RFID passive tags are distributed in an area of interest and attached to mobile objects. All the tags have the same resources and computational capabilities. The passive tag has no power source and gets activated based on the electromagnetic waves that are sent from the reader at the beginning of the communication. The RFID reader is a scanning device that is either in a fixed position or mobile handheld. It has more resources and computational capability than the passive tag. It collects the tag information such as the electronic product code (EPC) [10], which is a 96-bit string

of data that contains the tag identity, organization, protocol, product type, and owner. The reader reports the scanned information to the database server. The server is a centralized database device with a computer program that delivers, stores, and manages all the information of the readers and tags. The reader interrogates the tag in the range by sending a challenging request signal to start the communication. The tag responds to the reader's request based on the approved protocol to verify its legitimate identity. The reader forwards the tag's response to the server to search for the correct information of the tag in the database. The server supports the reader to authenticate the tag to start a secure channel between the reader and the tag for their further communication. Additionally, the tag uses the approved protocol to verify the reader's identity to avoid compromising the secret information or location of the tag.

*3.2. Serverless Model*

The server role is eliminated in the proposed serverless model of RFID. The backend server is not available during the communication between the reader and the tag. The reader and tag should be able to verify each other and process the authentication messages successfully while the server is offline. Because the passive tag is considered a low constraint device with scarce resources, the transmitted message between the reader and the tag should carry simple operations within the capability of the tag to perform. Therefore, we consider the elliptic curve cryptography that can be operated by the passive tag to exchange the secret keys. We employed the elliptic curve key agreement based on the discrete log problem in the Diffie–Hellman algorithm [23], which allows the reader and the tag to establish a shared key from their own public and private keys through an insecure channel to encrypt the transmitted messages. The elliptic curve is a plane curve over a finite field that contains points satisfying the following equation:

$$y^2 = x^3 + ax + b \tag{1}$$

The protocol uses the multiplicative group of integers modulo $P$, and $G$ as a primitive root modulo $P$, where $P$ is prime. The reader and the tag choose random integers **a** and **b**, respectively, as their private keys and compute their public keys as the following:

$$A = G^a \bmod P \tag{2}$$

$$B = G^b \bmod P \tag{3}$$

The values of $A$ and $B$ are exchanged between the reader and the tag. Then, the reader computes the shared secret $s$ using the received $B$ and $P$ as the following:

$$s = B^a \bmod P \tag{4}$$

The tag also computes the shared secret $s$ using the received $A$, and $G$, $P$ as the following:

$$s = A^b \bmod P \tag{5}$$

As a result, both the reader and the tag end up calculating the same value as their shared secret keys because the modulo rules satisfy the following:

$$A^b \bmod P = G^{ab} \bmod P = G^{ba} \bmod P = B^a \bmod P \tag{6}$$

which also means:

$$(G^a \bmod P)^b \bmod P = (G^b \bmod P)^a \bmod P \tag{7}$$

Based on the points $P$ and $G$, the resulted shared secret can take any value between **1** and **P-1** that satisfies the following condition: $1 \leq s \leq P - 1$. The security of the elliptic curve algorithm lies in the complexity of computing the original values of public and private keys to obtain the secret key.

## 4. Proposed Protocol

The communication between the system entities in SLEC protocol has three phases; setup phase, authentication phase, and an optional recovery phase. The recovery phase is needed to renew the values in the reader and tag from a new server. Table 1 denotes the protocol notations.

**Table 1.** Protocol Notations.

| Symbol | Definition |
|---|---|
| $P$ | Point generator of G |
| $G$ | An additive group of prime order $q$ on an elliptic curve |
| $y_{prv}, r_{prv}, t_{prv}$ | Private keys of server, reader, tag |
| $Y_{pub}, R_{pub}, T_{pub}$ | Public keys of server, reader, tag |
| $X_i$ | Tag identifier |
| $Tag_i$ | tag ID, group ID, Timestamp |
| $Gk$ | Group ID |
| $List_k$ | List of tags share the same group ID |

### 4.1. Setup Phase

This phase handles transferring the necessary data and values from the server to the reader and the tag. The server, the reader, and the tag share by the manufacturer: the elliptic curve point generator and the server public key. The tag by default stores its own random identifier that is updated every session to protect the real identity of the tag. The setup phase is also considered a renewal phase, such that the reader and the tag request new values to start a new communication session. The renewal phase is necessary when the timestamp expires, or any secret value is compromised to an unauthorized party.

Unlike the currently available RFID protocols, the setup phase environment in SLEC protocol is assumed to be insecure and functions as the following steps:

1. The reader and the tag generate random numbers $r_{prv1}$, $t_{prv1}$, respectively, where $r, t \ \epsilon Z_q$, then compute their public key using the private keys and the point generator as:

$$R_{pub1} = r_{prv1} * P \tag{8}$$

$$T_{pub1} = t_{prv1} * P \tag{9}$$

2. The tag and the reader compute the server secret message of $M_1$, $M_2$, respectively, using their private keys $t_{prv1}$, $r_{prv1}$ and the stored public key of the server $Y$ as the following:

$$M_1 = r_{prv1} * Y \tag{10}$$

$$M_2 = t_{prv1} * Y \tag{11}$$

3. The reader and tag send the computed server shared secret $M_1$, $M_2$ to the server, then the server obtains the public keys of both reader and tag as:

$$R'_{pub1} = y^{-1} * M_1 \tag{12}$$

$$t'_{pub1} = y^{-1} * M_2 \tag{13}$$

4. The server, in turn, computes the shared secret $M_1$, and $M_2$ for each reader and tag using the server private key $y_{prv}$ for further communication with the reader $R$ and the tag $T$ as the following:

$$M'_1 = y_{prv} * R == r_{prv1} * Y = M_1 \tag{14}$$

$$M'_2 = y_{prv} * T == t_{prv1} * Y = M_2 \tag{15}$$

5.  The server generates and stores the following information for each tag:

    - Random Xi as a tag identifier
    - Group ID Gk
    - Timestamp Ts

    such that $Tag_i = \{X_i, Gk, Ts\}$

6.  The server then generates a list of tags for each reader, which contains a group of tags that share the same group ID. The group ID will have multiple advantages in our proposed system. It provides security against tracing a specific tag when the reader receives multiple signals from different tags. It also reduces the communication signals in the network and the number of transmitted messages. Further, it reduces the computation overhead on the reader side when only the tags sharing the same group ID respond to the reader's request.

7.  Further, the server sends the reader the tag list as $M_3$, and sends the tag data to the tag as $M_4$.

$$M_3 = List_k + h\left( R'_{pub1}, M'_1 \right) \tag{16}$$

$$M_4 = Tag_i + h\left( T'_{pub1}, M'_2 \right) \tag{17}$$

8.  The reader computes and verifies the hash value to obtain the list of tags. The tag also verifies the hash value to obtain the tag information. The server current public key is not shared during the communication, so only the legitimate reader and tag that have the real server public key will be able to compute and verify the hash value to obtain the messages sent by the server.

*4.2. Authentication Phase*

When the setup phase is completed successfully, each reader will have a list of tags that have: a tag ID, group ID, timestamp for each tag, and the tag will have: a tag ID, group ID, and timestamp. The communication starts with mutual authentication between the reader and the tag, as demonstrated in the following steps:

1.  The reader generates a random number $r_{prv2}$, where $r \in Z_q$, then computes its public key using the private keys and the point generator as:

$$R_{pub2} = r_{prv2} * P \tag{18}$$

2.  The reader computes the $M_1$ and $M_2$ and sends them to the tag

$$M_1 = h\left( Gk_j \right) \tag{19}$$

$$M_2 = R_{pub2} \oplus Ts \tag{20}$$

3.  The tag processes five steps:

    - Validates $M_1 = h\left( Gk_j \right)$ using the current or old value of $Gk$ to verify the intended group. Based on the group verification, the tag generates a random number $t_{prv2}$, where $t \in Z_q$ and computes its own public key
    $$T_{pub2} = t_{prv2} * P \tag{21}$$

    - Obtains the reader public key from $M_2$

    $$R_{pub2} = (M_2 \oplus Ts) - Y \tag{22}$$

- Computes the secret share key with the reader using the reader obtained public key

$$M_3 = t_{prv2} * R_{pub2} \tag{23}$$

- Computes the authentication message $M_4$ and sends it to the reader

$$M_4 = h\left(X_i, R_{pub2}, T_{pub2}, Gk_j\right) \tag{24}$$

- Update the values of the tag $ID$, $Gk_j^{-1}$, and $Gk_j$

$$X_i = PRNG(X_i) \tag{25}$$

$$Gk_j{}^{old} = Gk_j \tag{26}$$

$$Gk_j = PRNG\left(Gk_j\right) \tag{27}$$

4. The reader extracts $T'_{pub2}$ from the received message and verifies the hash value of $M_4$ to authenticate the tag

$$T'_{pub2} = r_{prv2}^{-1} * M_3 \tag{28}$$

5. The reader updates the values of the tag $ID$, $Gk_j$, then computes $M_5$ and sends it to the tag

$$M_5 = h\left(X_i, R_{pub2}, T_{pub2}\right) \tag{29}$$

$$X_i = PRNG(X_i) \text{ (Equation (25))}$$

$$Gk_j = PRNG\left(Gk_j\right) \text{ (Equation (27))}$$

6. The tag verifies $M_5$ to authenticate the reader.

### 4.3. Recovery Phase

In an event where any value of the communication is compromised, the tag or the reader is able to renew the communication values from any server checkpoint during the transportation route. The recovery phase is similar to the secure setup phase presented earlier. The tag and reader will exchange their newly generated public keys using the server's public key stored in their memory. This will allow the reader and tag to be retrieved back into the network with new values.

In our SLEC protocol, we created an RFID network with a dynamic size so that the number of readers and tags can be increased or decreased. We included one server, five readers, and twenty tags that are placed in objects such as cars. The distance range between the tags and readers is initially assumed to be a few meters based on the reading range of electronic product code class1 generation 2 of RFID passive tags [10]. The server initializes a database table to store all the readers' and tags' unique IDs. The readers are placed in fixed positions such as poles along the route of the mobile tags. Before the car departs the dealership inventory, the setup phase is executed, and all the values are stored in the tag and readers. During the tag movement, the reader can scan the tag to perform the mutual authentication and thus, obtain the required information of the tag. The protocol is implemented using Python programming language. The protocol process is presented in Algorithm 1.

---

**Algorithm 1** SLEC (executed by server, reader, and tag in SLEC protocol)

---

1: *Setup Phase:*

2: **Input parameters:**

3:      minimum value for server_public_key ($Y$)

4:      point generator ($P$)

5:      tag random identifier ($X_i$) in server

6: *Reader:*

7: **for each** reader **do**

8:      Select random $r_{prv1} \; \epsilon Z_q$

9:      $R_{pub1} \; = r_{prv1} * P$

10:     $M_1 \; = r_{prv1} * Y$

11:     **for each** tag in TagList **do**

12:        Select random $t_{prv1} \; \epsilon Z_q$

13:        $T_{pub1} \; = t_{prv1} * P$

14:        $M_2 \; = t_{prv1} * Y$

15:        send $M_2$ to reader

16:     **end for**

17:     forward $M_1$ , $M_2$ to server

18: **end for**

19: *Server:*

20: extract $R_{pub1}$ , $T_{pub1}$ from $M_1$ , $M_2$

21: $R'_{pub1} = y^{-1} * M_1$

22: $T'_{pub1} = y^{-1} * M_2$

23: generate $Tag_i : [X_i, Ts, Gk_j]$

24: create $TagList_k : [Tag_i, .., Tag_n]$

25:     $M_3 = TagList_i + h(R'_{pub1}, M_1)$

26:     $M_4 = Tag_i + h(T'_{pub1}, M_2)$

27: send $M_3$ to reader

28: send $M_4$ to tag

29: *Reader:*

30: verify the hash value and extract $TagList_i = M_3 - h(R'_{pub1}, M_1)$

31: *Tag:*

32: verify the hash value and extract $Tag_i = M_4 - h(T'_{pub1}, M_2)$

33: *Authentication Phase:*

34: *Server is offline*

35: *Reader:*

36: **for each** reader **do**

37:     select random $r_{prv2} \; \epsilon Z_q$

38:     $R_{pub2} \; = r_{prv2} * P$

39:     $M_1 \; = h\big(Gk_j\big)$

40:     $M_2 = R_{pub2} \bigoplus Ts$

41:     send $M_1$ , $M_2$ to tag

42: **end for**

43: *Tag:*

44: **for each** tag **do**

45:     validate $M_1$ to verify the group

46:     **if** $M_1$ == True **do**

47:       select random $t_{prv2} \; \epsilon Z_q$

48:     $T_{pub2} \; = t_{prv2} * P$

49:       extract $R_{pub2} \; = M_2 \bigoplus Ts$

50:       $M_3 = t_{prv2} * R_{pub2}$

51:       $M_4 = h\big(X_i, R_{pub2} , T_{pub2} , Gk_j\big)$

52:       update:

53:         $X_i = PRNG(X_i)$

54:         $Gk_j^{-1} = Gk_j$

---

---

55:             $Gk_j = PRNG\big(Gk_j\big)$
56:             Send $M_3$ ,$M_4$ to reader $\leftarrow$
57:         **end if**
58: **end for**
59: *Reader:*
60: **for each** tag in TagList **do**
61:         extract $T'_{pub2} = r^{-1}_{prv2} * M_3$
62:         validate $M_4$ to authenticate tag
63:         **if** $M'_4 == M_4$ **do**
64:             Tag is authenticated
65:             $M_5 = h\big(X_i, R_{pub2} , T'_{pub2}\big)$
66:             update:
67:             $X_i = PRNG(X_i)$
68:             $Gk_j = PRNG\big(Gk_j\big)$
69:             send $M_5$ to tag
70:         **end if**
71: **end for**
72: *Tag:*
73: validate $M_5$ to authenticate reader
74: **if** $M'_5 == M_5$
75:     reader is authenticated
76: **end if**

---

## 5. Security Analysis

In this section, we present the system performance and security analysis of SLEC. We also compare the SLEC to other serverless protocols. The security of SLEC mainly depends on the public key of the central server, which is securely disseminated to all readers and tags. Further, the setup and authentication phases can then be executed in insecure networks while maintaining system requirements and defending security threats.

### 5.1. Analysis of System Requirements

The SLEC protocol maintains the system requirements that are necessary to create a secure and reliable RFID system, such as mutual authentication, confidentiality, integrity, privacy, forward secrecy, anonymity, and availability.

#### 5.1.1. Mutual Authentication

The protocol allows both the reader and the tag to perform a mutual authentication, since only the legitimate tag can extract the public key of the reader from the message $M_2$. Furthermore, only the legitimate reader can calculate the hash value in the message $M_5$ to prove its identity to the tag. As a result, mutual authentication is satisfied.

#### 5.1.2. Privacy and Confidentiality

The transmitted message is confidential because the authentication messages are secured by a hash value, which can only be computed by authorized entities using their secret keys. The privacy of the tag is satisfied as the secret information is protected and not transmitted in clear.

#### 5.1.3. Message Integrity

The message integrity factor is also satisfied because the messages are combined with a digital signature of the sender.

### 5.1.4. Forward and Backward Secrecy

The tag and reader generate new secret values in every authentication session to avoid tracking the previous or forward session or obtaining any valuable data. Thus, an adversary cannot perform a successful authentication from any past or expired sessions or anticipate the following authentication messages.

### 5.1.5. Anonymity

The EPC of the tag is not used in the protocol, but only the tag random identifier is used, which is updated every session. As a result, the private information stored in the tag is kept secret.

### 5.1.6. Availability

The protocol provides a recovery mechanism to maintain system availability. In an event where any tag or any secret value of the communication is compromised, the system can recover the tag by sending new values to the tag during the recovery phase to perform a new authentication session as long as the public key of the server remains secret. Otherwise, a new setup phase is required to feed the tag with a new public key for the server.

### 5.1.7. Scalability

We introducedd the concept of tag grouping in SLEC. We combined a number of tags into one group that shares the same group ID with all the tags, but different tag IDs. This mechanism allows the system to reduce the communication signals that are transmitted in the network, since only the tags with the same group ID will respond to the reader request. Moreover, the grouping mechanism reduces the computation overhead on the reader side when identifying a tag from a large number of tags. As a result, the protocol is scalable by maintaining a consistent operation overhead on both the reader and the tag sides.

Table 2 demonstrates the comparison of the system requirements that are satisfied in SLEC, the SAP protocol proposed by Mtita et al. [16], and the STS protocol proposed by Sundaresan et al. [3].

**Table 2.** Comparison of System Requirements.

| System Requirement | SAP | STS | SLEC |
|---|---|---|---|
| Mutual Authentication | Y | Y | Y |
| Privacy and Confidentiality | N | Y | Y |
| Message Integrity | Y | Y | Y |
| Forward and Backward Secrecy | N | N | Y |
| Anonymity | * | Y | Y |
| Availability | N | N | Y |
| Scalability | N | N | Y |

Y: satisfied; N: not satisfied; *: Not applicable.

### 5.2. Analysis of Security Requirements

The protocol is based on the Diffie–Hellman digital signature algorithm using a 256-bit key, which is equivalent to the RSA algorithm with a 3072-bit key that is longer than the commonly used key of 2048 [24]. This gives a higher level of security to the SLEC algorithm. Therefore, the protocol is secure against different security attacks that most of the RFID protocols can experience.

### 5.2.1. Replay Attack

The proposed SLEC is secure against replay attack, as the authentication session involves timestamps and freshly generated random values as private keys for both reader and tag. If an

adversary eavesdrops on the communication channel to replay the tag response, he will not be able to extract any message from the reader or the tag, and the timestamp will not match the current session.

**Lemma 1.** *SLEC is secure against replay attack.*

**Proof.** Adversary replays old session1 to the reader:

$$M_3 = t_{prv1} \times (r_{prv1} \times P)$$

$$M_4 = h\left(X_1 + \left(r_{prv1} \times P\right) + \left(t_{prv1} \times P\right) + Gk_1\right)$$

Reader verifies in session2:

$$T'_{pub2} = r_{prv2}^{-1} \times [t_{prv1} \times (r_{prv1} \times P)]$$

$$M'_4 = h(X_2 + \left(r_{prv2} \times P\right) + [r_{prv2}^{-1} \times t_{prv1} \times (r_{prv1} \times P)] + Gk_2)$$

Verification fails since $M'_4 \neq M_4$. Unauthorized tag is not authenticated. □

### 5.2.2. Man-In-The-Middle Attack

If an adversary interrupts the message transmitted by a reader or a tag to modify it and send it back as a real message, the message will not be extracted by any entity and the communication will be terminated if no response is sent because all the messages transmitted in the authentication session involve validating the values before extracting any data from them. Therefore, the SLEC protocol is resistant to Man-In-The-Middle attack.

**Lemma 2.** *SLEC is secure against modification.*

**Proof.** Adversary A intercept message 3 and 4 and modifies the tag information by A:

$$M_3 = a_{prv1} \times (r_{prv1} \times P)$$

$$M_4 = h\left(X_a + \left(r_{prv1} \times P\right) + \left(a_{prv1} \times P\right) + Gk_a\right)$$

Reader verifies in session2:

$$A'_{pub1} = r_{prv1}^{-1} \times [a_{prv1} \times (r_{prv1} \times P)]$$

$$M'_4 = h(X_1 + \left(r_{prv1} \times P\right) + [r_{prv1}^{-1} \times t_{prv1} \times (r_{prv1} \times P)] + Gk_1)$$

Verification fails since $M'_4 \neq M_4$. The tag id and group key used in the message sent by A are not the same in the reader list for the requested tag. Unauthorized tag is not authenticated. □

### 5.2.3. Traceability Attack

An adversary can trace the signals sent by a specific tag to identify the tag location. However, the reader in the SLEC protocol broadcasts the message signals to a group of tags that respond to the reader for the same message request. This will result in sending different signals from different locations to confuse the adversary from tracking a specific tag to obtain its location. Accordingly, the protocol is resistant to tracing.

**Lemma 3.** *SLEC is secure against tracing.*

**Proof.** To distinguish the difference between two tags $T_1$ and $T_2$, an adversary has to construct the correct hash value with correct tag id ($X$), timestamp ($Ts$), and group key ($Gk$), which are only transmitted during the setup phase:

$$M_4 = Tag_i + h\left[\left(t_{prv1} \times P\right) + \left(t_{prv1} \times (y \times P)\right)\right]$$

$$Tag_i = M_4 - h\left[\left(t_{prv1} \times P\right) + \left(t_{prv1} \times (y \times P)\right)\right]$$

The adversary has to solve the correct elliptic curve discrete logarithm problem (ECDLP) to obtain the secret values in $Tag_i$ that are used in the communication. □

### 5.2.4. Impersonate Attack

It is unlikely for any adversary to impersonate the reader or the tag in our protocol since they used a shared point generator algorithm (P) that is only known to the legitimate server, reader, and tag. Therefore, the adversary cannot compute the required messages to pass the authentication.

**Lemma 4.** *SLEC is secure against impersonation.*

**Proof.** Reader: $M_1 = h(Gk)$

$$M_2 = (r_{prv} \times P) \bigoplus Ts$$

Adversary: $a_{prv}, A_{pub} = a_{prv} \times P$

$$M_{3a} = a_{prv} \times (ra_{prv} \times P)$$

$$M_{4a} = h\left(X_a + \left(ra_{prv} \times P\right) + A_{pub} + Gk_a\right)$$

Reader: $A'_{pub} = r_{prv}^{-1} \times \left[a_{prv} \times ra_{prv} \times P\right)]$

$$M_4 = h\left(X + R'_{pub} + A_{pub} + Gk\right)$$

Validation fails since $M_4 \neq M_{4a}$. Unauthorized tag is not authenticated. □

### 5.2.5. Desynchronization Attack

The tag in SLEC stores the new and previous values of the group identifier that is used at the beginning of the authentication phase. This allows the tag to authenticate the reader if the previous session was interrupted by an adversary to break the synchronization. The communication values are also updated after every successful authentication session using the same algorithm and inputs, to maintain the synchronization state between the network entities.

### 5.2.6. Denial of Service Attack

In SLEC, the tag and reader generate their keys separately using the same key generation algorithm, so there is no synchronous update of the keys between the server and the tag for the attack to occur.

Table 3 demonstrates the comparison of the security attacks resistance between our SLEC protocol, the SAP protocol proposed by Mtita et al. [16], and the STS protocol proposed by Sundaresan et al. [3].

**Table 3.** Comparison of Security Resistance.

| Attacks | SAP | STS | SLEC |
|---|---|---|---|
| Replay Attack | Y | Y | Y |
| Man-in-the-Middle | * | * | Y |
| Eavesdropping | Y | * | Y |
| Impersonate Attack | Y | Y | Y |
| Traceability Attack | Y | Y | Y |
| Desynchronization | * | Y | Y |
| Denial of Service | * | Y | Y |

Y: Satisfied; N: Not satisfied; *: Not applicable.

### 5.3. Analysis of Computation Cost

Because the passive tag used in the RFID system has limited capabilities and resources, it is crucial to consider the computation and security features for the appropriate application. Even though the elliptic curve has higher computation overhead on both the reader and the tag, we provide a higher security level in our SLEC protocol that satisfies the resistance to all the security attacks. Moreover, we compare our protocol with additional server-based elliptic curve protocols, such as the IECC protocol proposed by Farash [11] and the EECC protocol proposed by Zhang and Qi [12] to illustrate a well-defined measurement for the computation complexity. The comparison shows that there is no significant additional cost between the previously proposed ECC-based protocols and our SLEC protocol, although SLEC is completely serverless in the authentication phase. Table 4 demonstrates the operations computed by the tag and the number of transmitted messages from the reader and the tag during the authentication phase.

**Table 4.** Comparison of computation cost.

| Protocol | Operation | Tag | Reader |
|---|---|---|---|
| SAP [16] | $2T_H + 2T_{RNG}$ | 1 | 2 |
| STS [3] | $7T_{XOR} + 3T_{PRNG}$ | 1 | 1 |
| IECC [11] | $2T_{SMUL} + 2T_H$ | 1 | 2 |
| EECC [12] | $2T_{SMUL} + T_{SAD} + 2T_H$ | 1 | 2 |
| SLEC | $2T_{SMUL} + 3T_H$ | 1 | 2 |

$T_{SMUL}$: scalar multiplication; $T_{SAD}$: scalar addition; $T_H$: oneway hash; $T_{XOR}$: XOR; and $T_{PRNG}$: pseudo-random number generation.

### 5.4. Analysis of Communication Cost

In this section, we analyze the communication cost of the proposed protocol and compare it with other ECC-based protocols to show the improvement of the communication cost using ECC.

We use a standard 163-bit NIST elliptic curve using 5 MHz tags. We will emphasize the elliptic curve operations in our comparison. The running time of scalar multiplication in ECC is 64 ms. In our proposed protocol, we require two scalar multiplications that are executed in $64 \times 2 = 128$ ms. We use an elliptic curve defined over a finite field of $F(2^{163})$. We assume the length of the elliptic curve element is 163 bits, the elliptic curve point is 42 bytes, and the output of the hash function is 20 bytes. The comparison results of the performance are depicted in Table 5. According to the performance results, the proposed protocol shows slightly better performance and communication cost that is more suitable for real implementation.

**Table 5.** Comparison of communication cost.

| Protocol | From Tag | From Reader | Total |
|----------|----------|-------------|-------|
| IECC [11] | 124 | 62 | 186 |
| EECC [12] | 124 | 41 | 165 |
| SLEC | 62 | 86 | 148 |

Point = 42 bytes, Element = 21 bytes, Hash output = 20 bytes, ID = 4 bytes, Timestamp = 4 bytes.

## 6. Formal Verification

We have conducted a formal verification for SLEC to prove the correctness of the algorithms used. ProVerif [25] is a powerful tool used to analyze the security of cryptographic protocols. It is an automatic cryptographic protocol verifier that is developed by Bruno Blanchet to validate the security and authentication properties of the cryptographic algorithms in formal models.

In this section, we use the ProVerif tool to validate reachability and secrecy (security) and correspondence assertion (authentication) of SLEC protocol. The results of the verification process are also presented.

### 6.1. Adversary Model

The ProVerif verification is based on a model where the adversary can intercept, alter, and inject the messages into an insecure network. In SLEC, the adversary has initial knowledge of the finite set of parameters that increase during the protocol execution in parallel with the adversary. No matter how the adversary interacts with the protocol, ProVerif verifies the secrecy of the messages and values transmitted between the server, the reader, and the tag. Therefore, the secret messages will never be a part of the adversary knowledge to run the protocol successfully. The results of the ProVerif verification in this section show that the protocol preserves the secrecy of the messages and values.

### 6.2. Reachability and Secrecy

ProfVerif Reachability and Secrecy analyzes the security properties of the protocol against multiple types of attacks. We investigate the reachability of a term $x$ by an adversary $A$, so we assess the secrecy of $x$ with respect to the modeled protocol. In SLEC, we use ProVerif to test whether the secret messages in the setup phase "Ms", and the secret messages in the authentication phase "Ma" are not available to an adversary $A$. We represent the messages transmitted in the setup phase from the server, the reader, and the tag as "Mss", "Msr", and "Mst", respectively. Moreover, we represent the messages transmitted in the authentication phase from the reader and the tag as "Mar" and "Mat", respectively. The complete verification process is demonstrated in Figure A1 in Appendix A. The results of the verification process conclude, "RESULT not attacker(Mst1[]) is true", which means the setup phase message $M_1$ from the tag is unreachable, and an attack cannot be conducted against the protocol successfully. Similarly, "RESULT not attacker(Mar1[]) is true" means the authentication phase message $M_1$ from the reader is unreachable and secure against the attacks. All setup phase and authentication phase messages were tested and resulted in true reachability and secrecy proof.

### 6.3. Correspondence Assertion

The correspondence assertion in ProVerif is to model the authentication of the protocol using a sequence of events. We apply a sequence of events to verify the authentication of the reader to the tag and the authentication of the tag to the reader through the encrypted messages individually. The complete verification process of the authentication is presented in Figure A2 in Appendix A. The results of the correspondence assertion verification show "RESULT inj-event(termReader(x)) ==> inj-event(acceptsReader(x)) is true", which means the reader is authenticated by the tag, and "RESULT inj-event(termTag(x_24)) ==> inj-event(acceptsTag(x_24)) is true", which means the tag is

authenticated by the reader. The verification results confirm that SLEC protocol achieves a successful mutual authentication between the reader and the tag.

## 7. Conclusions

RFID is the new alternative to physical barcodes, which is widely being used for products inventory and asset tracking. Serverless RFID protocols are being developed to provide a dynamic network, so that mobile tags (attached to items) can be searched and identified in different locations away from the server. Numerous research considers the issue of system security and data privacy in the reader–tag mutual authentication. As an RFID network carries along sensitive information, and passive tags have limited resources, many security protocols have been implemented at a minimal computational cost using simple operations. However, this does not make it secure against cyberattacks. We propose a secure serverless RFID protocol (SLEC) that uses ECC, which is based on the Diffie–Hellman encryption algorithm. ECC is classified as a public-key encryption algorithm that low constraint devices, such as RFID passive tags, can handle efficiently. We have proved that SLEC is secure against all attacks. The reader in SLEC protocol is completely capable of identifying and authenticating mobile tags without the need for a server. We also introduced the tag grouping mechanism to reduce the reader's computation overhead due to ECC computation, while identifying a tag in a large-scale network. Tag grouping provides a scalable system that is not affected by the tag population size. Furthermore, SLEC protocol has a recovery mechanism to renew compromised values by any server in the network. The protocol was tested using ProVerif, which is a cryptographic verification tool. ProfVerif proved that SLEC achieves secure authentication. As for future work, we will work on implementing a testbed for the protocol using industrial passive tags and readers.

**Author Contributions:** Conceptualization, R.B., and A.A.; methodology, R.B.; software, R.B.; validation, R.B., and A.A.; formal analysis, R.B. and A.A.; writing—original draft preparation, R.B.; writing—review and editing, A.A.; supervision, A.A.; project administration, A.A.

**Funding:** This research received no external funding.

**Conflicts of Interest:** The authors declare no conflicts of interest.

## Appendix A

| Reachability and Secrecy |
|---|

Process:

{1}new p: P;

{2}new y_22: Y;

{3}new tprv1: pu_pr_key;

{4}new rprv1: pu_pr_key;

{5}new xi: TagID;

{6}new Gxi: GK;

(

    {7}!

    {8}let Tpub1: pu_pr_key = find_R_and_Tpubs(p,tprv1) in

    {9}let Mst1_23: pu_pr_key = setPhase_encrypt(tprv1,y_22) in

    {10}out(ch, Mst1_23);

    {11}in(ch, Mss4': Tag_and_Hash);

    {12}let tag': Tag = verify_tag(Mss4',h(Tpub1,Mst1_23)) in

    {13}in(ch, ms1: xored_key);

    {14}let Ms1': GK = validate_Ms1(ms1) in

    {15}new ggkk: GK;

    {16}if (Ms1' = ggkk) then

    {17}new tprv2: pu_pr_key;

    {18}let Tpub2: pu_pr_key = find_R_and_Tpubs(p,tprv2) in

    {19}in(ch, ms2: pu_pr_key);

    {20}new ts2': timeStampes;

    {21}let Rpub2': pu_pr_key = xor(ms2,ts2') in

    {22}let Mat3_24: pu_pr_key = authen_phase_encrypt(tprv2,Rpub2') in

    {23}out(ch, Mat3_24);

    {24}new xi_25: TagID;

    {25let Mat4_26: Hashing = h_ph2_2(xi_25,Rpub2',Tpub1,Ms1') in

    {26}out(ch, Mat4_26);

    {27}let xxi: TagID = PRNG(xi_25) in

    {28}let GKi: GK = PRNG_Group(ggkk) in

    {29}in(ch, Mar5': Hashing);

    {30}let Mar55: Hashing = h_ph2_3(xi_25,Tpub2,Rpub2') in

    {31}if (Mar55 = Mar5') then

    0

**Figure A1.** *Cont*.

```
  ) | (
    {32}!
    {33}let Rpub1: pu_pr_key = find_R_and_Tpubs(p,rprv1) in
    {34}let Msr2_27: pu_pr_key = setPhase_encrypt(rprv1,y_22) in
    {35}out(ch, Msr2_27);
    {36}in(ch, Mss3': TagList_and_Hash);
    {37}let taglisht': TageList = verify_tagList(Mss3',h(Rpub1,Msr2_27)) in
    {38}new rprv2: pu_pr_key;
    {39}let Rpub2: pu_pr_key = find_R_and_Tpubs(p,rprv2) in
    {40}new GKii: GK;
    {41}let Mar1_28: hash_GK = h_ph2_(GKii) in
    {42}out(ch, Mar1_28);
    {43}new ts2: timeStampes;
    {44}let Mar2_29: pu_pr_key = xor(Rpub2,ts2) in
    {45}out(ch, Mar2_29);
    {46}in(ch, Mat3': pu_pr_key);
    {47}let Tpub2': pu_pr_key = authen_phase_dencrypt(Mat3',re_pu_pr_key(rprv2)) in
    {48}in(ch, Mat4': Hashing);
    {49}new xii': TagID;
    {50}let Mar5: Hashing = h_ph2_3(xii',Rpub2,Tpub2') in
    {51}out(ch, Mar5)
) | (
    {52}!
    {53}in(ch, (Mst1': pu_pr_key,Msr1': pu_pr_key));
    {54}let Tprv1': pu_pr_key = setPhase_decrypt(Mst1',getYinv(y_22)) in
    {55}let Rprv1': pu_pr_key = setPhase_decrypt(Msr1',getYinv(y_22)) in
    {56}new ts: timeStampes;
    {57}let tagi: Tag = create_tag(xi,ts,Gxi) in
    {58}let tagListi: TageList = create_taglist(tagi) in
    {59}let Mss3_30: TagList_and_Hash = prepare_tagList(tagListi,h(Rprv1',Msr1')) in
    {60}out(ch, Mss3_30);
    {61}let Mss4_31: Tag_and_Hash = prepare_tag(tagi,h(Tprv1',Mst1')) in
    {62}out(ch, Mss4_31)
)
-- Query not attacker(Mst1[])
Completing...
Starting query not attacker(Mst1[])
RESULT not attacker(Mst1[]) is true.
-- Query not attacker(Msr2[])
Completing...
```

**Figure A1.** *Cont.*

Starting query not attacker(Msr2[])

RESULT not attacker(Msr2[]) is true.

-- Query not attacker(Mss3[])

Completing...

Starting query not attacker(Mss3[])

RESULT not attacker(Mss3[]) is true

-- Query not attacker(Mss4[])

Completing...

Starting query not attacker(Mss4[])

RESULT not attacker(Mss4[]) is true.

-- Query not attacker(Mar1[])

Completing...

Starting query not attacker(Mar1[])

RESULT not attacker(Mar1[]) is true.

-- Query not attacker(Mar2[])

Completing...

Starting query not attacker(Mar2[])

RESULT not attacker(Mar2[]) is true.

-- Query not attacker(Mat3[])

Completing...

Starting query not attacker(Mat3[])

RESULT not attacker(Mat3[]) is true.

-- Query not attacker(Mat4[])

Completing...

Starting query not attacker(Mat4[])

RESULT not attacker(Mat4[]) is true.

**Figure A1.** Verification results of reachability and secrecy.

---

Correspondence Assertion

Process:

{1}new p: P;

{2}new y_22: Y;

{3}new tprv1: pu_pr_key;

{4}new rprv1: pu_pr_key;

{5}new xi: TagID;

{6}new Gxi: GK;

(

    {7}!

    {8}let Tpub1: pu_pr_key = find_R_and_Tpubs(p,tprv1) in

    {9}let Mst1_23: pu_pr_key = setPhase_encrypt(tprv1,y_22) in

    {10}out(ch, Mst1_23);

    {11}in(ch, Mss4': Tag_and_Hash);

    {12}let tag': Tag = verify_tag(Mss4',h(Tpub1,Mst1_23)) in

    {13}in(ch, ms1: xored_key);

    {14}let Ms1': GK = validate_Ms1(ms1) in

    {15}new ggkk: GK;

    {16}if (Ms1' = ggkk) then

    {17}new tprv2: pu_pr_key;

    {18}let Tpub2: pu_pr_key = find_R_and_Tpubs(p,tprv2) in

    {19}in(ch, ms2: pu_pr_key);

    {20}new ts2': timeStampes;

    {21}let Rpub2': pu_pr_key = xor(ms2,ts2') in

    {22}let Mat3_24: pu_pr_key = authen_phase_encrypt(tprv2,Rpub2') in

    {23}out(ch, Mat3_24);

    {24}new xi_25: TagID;

    {25}let Mat4_26: Hashing = h_ph2_2(xi_25,Rpub2',Tpub1,Ms1') in

    {26}out(ch, Mat4_26);

    {27}let xxi: TagID = PRNG(xi_25) in

    {28}let GKi: GK = PRNG_Group(ggkk) in

    {29}in(ch, Mar5': Hashing);

    {30}let Mar55: Hashing = h_ph2_3(xi_25,Tpub2,Rpub2') in

    {31}if (Mar55 = Mar5') then

    0

) | (

    {32}!

    {33}let Rpub1: pu_pr_key = find_R_and_Tpubs(p,rprv1) in

    {34}let Msr2_27: pu_pr_key = setPhase_encrypt(rprv1,y_22) in

    {35}out(ch, Msr2_27);

---

**Figure A2.** *Cont.*

```
    {36}in(ch, Mss3': TagList_and_Hash);

    {37}let taglisht': TageList = verify_tagList(Mss3',h(Rpub1,Msr2_27)) in

    {38}new rprv2: pu_pr_key;

    {39}let Rpub2: pu_pr_key = find_R_and_Tpubs(p,rprv2) in

    {40}new GKii: GK;

    {41}let Mar1_28: hash_GK = h_ph2_(GKii) in

    {42}out(ch, Mar1_28);

    {43}new ts2: timeStampes;

    {44}let Mar2_29: pu_pr_key = xor(Rpub2,ts2) in

    {45}out(ch, Mar2_29);

    {46}in(ch, Mat3': pu_pr_key);

    {47}let Tpub2': pu_pr_key = authen_phase_dencrypt(Mat3',re_pu_pr_key(rprv2)) in

    {48}in(ch, Mat4': Hashing);

    {49}new xii': TagID;

    {50}let Mar5: Hashing = h_ph2_3(xii',Rpub2,Tpub2') in

    {51}out(ch, Mar5)

) | (

    {52}!

    {53}in(ch, (Mst1': pu_pr_key,Msr1': pu_pr_key));

    {54}let Tprv1': pu_pr_key = setPhase_decrypt(Mst1',getYinv(y_22)) in

    {55}let Rprv1': pu_pr_key = setPhase_decrypt(Msr1',getYinv(y_22)) in

    {56}new ts: timeStampes;

    {57}let tagi: Tag = create_tag(xi,ts,Gxi) in

    {58}let tagListi: TageList = create_taglist(tagi) in

    {59}let Mss3_30: TagList_and_Hash =

prepare_tagList(tagListi,h(Rprv1',Msr1')) in

    {60}out(ch, Mss3_30);

    {61}let Mss4_31: Tag_and_Hash = prepare_tag(tagi,h(Tprv1',Mst1')) in

    {62}out(ch, Mss4_31)

)

-- Query not attacker(Mst1[])

Completing...

Starting query not attacker(Mst1[])

RESULT not attacker(Mst1[]) is true.

-- Query not attacker(Msr2[])

Completing...

Starting query not attacker(Msr2[])

RESULT not attacker(Msr2[]) is true.

-- Query not attacker(Mss3[])

Completing...
```

**Figure A2.** *Cont.*

Starting query not attacker(Mss3[])

RESULT not attacker(Mss3[]) is true.

-- Query not attacker(Mss4[])

Completing...

Starting query not attacker(Mss4[])

RESULT not attacker(Mss4[]) is true.

-- Query not attacker(Mar1[])

Completing...

Starting query not attacker(Mar1[])

RESULT not attacker(Mar1[]) is true.

-- Query not attacker(Mar2[])

Completing...

Starting query not attacker(Mar2[])

RESULT not attacker(Mar2[]) is true.

-- Query not attacker(Mat3[])

Completing...

Starting query not attacker(Mat3[])

RESULT not attacker(Mat3[]) is true.

-- Query not attacker(Mat4[])

Completing...

Starting query not attacker(Mat4[])

RESULT not attacker(Mat4[]) is true.

**Figure A2.** Verification results of correspondence assertion.

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
