# Peer review of "SLEC: A Novel Serverless RFID Authentication Protocol Based on Elliptic Curve Cryptography"

_electronics, doi:10.3390/electronics8101166_

Round 1

Reviewer 1 Report

The paper presents an authentication mechanism for RFID tags using ECC (Elliptic Curve Cryptography). I appreciate the idea and I want to mention that the problem to be solved by the paper is very challenging and relevant to the current state of the art.

I have some concerns and comments regarding the paper contents.

Usually, the RFID reader connects to RFID middleware (software) which is connected to the database, so there is no direct connection between the reader and the database. I suggest the authors to add the RFID middleware in figure 1. 

The authors present the serverless authentication algorithm and they state that passive RFID tags are used and that they can do the necessary computing. Also, some data is written on the tag, so the reader is actually tag writing capable device. Can you please clarify there aspects? I strongly recommend increasing the clarity of the presentation in the aspects regarding the functioning of the RFID system.

It would also be very interesting if the system was tested in a experimental setup. I understand that the proposed algorithm was formally verified using  ProVerif, but I wonder if the practical implementation of the system is feasible.

Although, the algorithm presentation is clear and the verification part is strong in my opinion. 

I recommend the authors to improve the presentation of RFID system.

Author Response

Thank you so much for the review. Please find the attached file.

Reviewer 2 Report

Line 295, equation 14: y_prv is undefined, is it servers private key? Line 305:M_3 = list_k + h(R_pub1, M_1) Does M3 send as plaintext? Also, h(R_pub1, M_1) only declare one thing: The server got the readers  public key.  I think the server should protect List_k, too. Line 306: M4 = Tag_i + + h(T;_pub1, M2). There are two plus signs, is it a mistake? Also, same as above, Tag_i is no relationship with h(T;_pub1, M2). An attacker can intercept the message, and send a fake Tag_i to a tag. The tag does not know which reader is authorized to access it. Line 320: M1 = h(Gk_j). Actually, an attacker can capture Gk_j since its sent as plaintext in M4 (line 306, eq 17). That is, the validating is meaningless. Its very easy to break the protocol:

Line 320: M1 = h(Gk_j).

Actually, an attacker can capture Gk_j since its sent as plaintext in M4 (line 306, eq 17). That is, the validating is meaningless.

Line 321: M2 = R_pub2 XOR T_s.  The attacker can use his own public key to replace R_pub2 because he knows T_s from M4 in equation 17.

Line 334: X_i = PRNG(X_i), GK_j old = GJ_j, GK_j = PRNG(GK_j). (Equation 25/26/27)

  The attacker can calculate the new X_i and GK_j because he knows the old values from M4 in equation 17. That is, the tag will authenticate the attacker instead of an authorized reader in Line 344 (equation 27).

Author Response

(The authors gave the same response as above.)

Reviewer 3 Report

The authors of this work propose a serverless protocol for RFID authentication. Such a protocol is based on Elliptic Curves to provide advanced security in the communication between the server, readers and tags. Once the protocol is described, it is tested using the ProVerif tool. 

The paper is well-written and I like the idea, but there are a couple of things that I think must be addressed before the paper gets published. 

I think that the related work section could include a table (similar to the tables included in later sections) that summarizes the previous proposals, advantages and disadvantages so the purpose of the SLEC protocol is clearer. 

The concept of Group ID is mentioned in the setup phase, however, it is not specified why it is required until later.

In section 4.1 it is said that the manufacturer assigns the elliptic curve point generator and server public key to the server, reader and tag. During the setup, the server generates a Taglist. Since the Tag of each element is updated in the authentication phase, it is not clear to me why the TagList in the server is not updated. At some point, the reader should check with the server that maintains the database the state of Tags and if it the server does not update the Taglist, I don't know how it will do it.

In line 354, authors say that they have included one server, five readers and twenty tags. Is that for the tests or for the simulation or for which reason?

In order to generate some messages, it is required to compute a hash function. How is that function specified?

Authors assume that only the server is vulnerable to Denial of Service Attacks, but what about the readers?

When the computation cost is estimated, the numbers in the Tag and Reader columns are not clear. Besides, the SLEC protocol updates the Tag identifier using a PRNG function that is not included in table 4, why?

In my opinion Figures 3 and 4 can be moved to an appendix because they can confuse a reader not familiar with the tool and only the outputs of the tool are relevant to prove the authors' points.

I also believe that it will be useful to make the code implementing the protocol available.

There are a couple of English spelling errors that could be easily corrected with a proofreading.

Author Response

(The authors gave the same response as above.)

Round 2

Reviewer 1 Report

The paper presents an authentication mechanism for RFID tags using ECC (Elliptic Curve Cryptography). I appreciate the idea and I want to mention that the problem to be solved by the paper is very challenging and relevant to the current state of the art.

It is not clear how the RFID passive tags can be used for intensive computing needed for the ECC.  Also, some data is written on the tag, so the reader is actually tag writing capable device. Can you please clarify there aspects?

I consider that the proposed scheme should be confirmed by experimental results which should be presented in the paper.

The paper quality should be improved.

Author Response

Thank you so much for your great review and comments. Please find attached our responses to your questions.

Thanks again.

Reviewer 2 Report

I have no more questions.

Author Response

Thank you so much for your review. We have read proof the article again as per your request. We truly appreciate all your comments and time.

Sincerely,

~